# Augmentative Topology Agents For Open-ended Learning

## Abstract

In this work, we tackle the problem of open-ended learning by introducing a method that simultaneously evolves agents and increasingly challenging environments. Unlike previous open-ended approaches that optimize agents using a fixed neural network topology, we hypothesize that generalization can be improved by allowing agents' controllers to become more complex as they encounter more difficult environments. Our method, Augmentative Topology EPOET (ATEP), extends the Enhanced Paired Open-Ended Trailblazer (EPOET) algorithm by allowing agents to evolve their own neural network structures over time, adding complexity and capacity as necessary. Empirical results demonstrate that ATEP results in general agents capable of solving more environments than a fixed-topology baseline. We also investigate mechanisms for transferring agents between environments and find that a species-based approach further improves the performance and generalization of agents.

## 1 Introduction

Machine learning has successfully been used to solve numerous problems, such as classifying images (Krizhevsky et al., 2012), writing news articles (Radford et al., 2019; Schick & Schütze, 2021) or solving games like Atari (Mnih et al., 2015) or chess (Silver et al., 2018). While impressive, these approaches still largely follow a traditional paradigm where a human specifies a task that is subsequently solved by the agent. In most cases, this is the end of the agent's learning—once it can solve the required task, no further progression takes place.

Through the motivation by the fact that humans have always learnt and innovated in an open-ended manner, *Open-ended learning* research field emerged (Stanley et al., 2017). For instance, humans did not invent microwaves to heat food, but to study radars. Vacuum tubes and electricity was invented for very different reason but we stumbled upon computers through them (Stanley, 2019).

In perspective of agent, Open-ended learning is a research field that rather than converge to a specific goal, the aim is to obtain an increasingly growing set of diverse and interesting behaviors (Stanley et al., 2017). One approach is to allow both the agents, as well as the environments, to change, evolve and improve over time (Brant & Stanley, 2017; Wang et al., 2019). This has the potential to discover a large collection of useful and reusable skills (Quessy & Richardson, 2021), as well as interesting and novel environments (Gisslén et al., 2021). Open-ended learning is also a much more promising way to obtain truly general agents than the traditional single task-oriented paradigm (Team et al., 2021).

The concept of open-ended evolution has been a part of artificial life (ALife) research for decades now, spawning numerous artificial worlds (Ray, 1991; Ofria & Wilke, 2004; Spector et al., 2007; Yaeger & Sporns, 2006; Soros & Stanley, 2014). These worlds consist of agents with various goals, such as survival, predation, or reproduction. Recently, open-ended algorithms have received renewed interest (Stanley, 2019), with Stanley et al. (2017) proposing the paradigm as a path towards the goal of human-level artificial intelligence.

A major breakthrough in open-ended evolution was that of *NeuroEvolution of Augmenting Topologies* (NEAT) (Stanley & Miikkulainen, 2002), which was capable of efficiently solving complex reinforcement learning tasks. Its key idea was to allow the structure of the network to evolve alongside the weights, starting with a simple network and adding complexity as the need arises. This inspired

future research about open-endedly evolving networks indefinitely (Soros & Stanley, 2014). Specifically, novelty search (Lehman et al., 2008), used the idea of *novelty* to drive evolution, instead of traditional objective-based techniques. This in turn led to the emergence of quality diversity (QD) algorithms (Lehman & Stanley, 2011; Mouret & Clune, 2015; Ecoffet et al., 2019; Nilsson & Cully, 2021), which are based on combining novelty with an objective sense of progress, where the goal is to obtain a collection of diverse and high-performing individuals.

While QD has successfully been used in numerous domains, such as robotic locomotion (Cully et al., 2015; Mouret & Clune, 2015; Tarapore et al., 2016), video game playing (Ecoffet et al., 2019) and procedural content generation (Khalifa et al., 2018; Earle et al., 2022), it still is not completely open-ended, where completely open-ended means to run indefinitely and create novel artifacts. One reason for this is that the search space for phenotypical behavior characteristics (or behavioral descriptors) remains fixed (Mouret & Clune, 2015). A second reason is that in many cases, the environment remains fixed, which limits the open-endedness of the algorithm (Wang et al., 2019). A way to circumvent this is to co-evolve problems and solutions, as is done by Minimal Criterion Coevolution (MCC) (Brant & Stanley, 2017). This co-evolutionary pressure allowed more complex mazes to develop, and better agents to solve them emerged, giving rise to an open-ended process.

However, MCC had some limits; for instance, it only allows new problems if they are solvable by individuals in the current population. This leads to only slight increases in difficulty, and complexity which only arises randomly. Taking this into account, *Paired Open-ended Trailblazer* (POET) (Wang et al., 2019) builds upon MCC, but instead allows the existence of unsolvable environments, if it was likely that some individuals could quickly learn to solve these environments. POET further innovates by transferring agents between different environments, to increase the likelihood of solving hard problems. While POET obtained state of the art results, its diversity slows down as it evolves for longer. Enhanced POET (Wang et al., 2020) adds improved algorithmic components to the base POET method, resulting in superior performance and less stagnation. Enhanced POET, however, uses agents with fixed topology neural network controllers. While this approach works well for simple environments, it has an eventual limit on the complexity of tasks it can solve: at some point of complexity, the fixed topology agents may not have sufficient capacity to solve the environments.

To address this issue, we propose *Augmentative Topology Enhanced POET* (ATEP), which uses NEAT to evolve agents with variable, and potentially unbounded, network topologies. We argue that fixed-topology agents will cease to solve environments after a certain level of complexity and empirically show that ATEP outperforms Enhanced POET (EPOET) in a standard benchmark domain. Finally, we find that using NEAT results in improved exploration and better generalization compared to Enhanced POET.

## 2 RELATED WORK

POET (Wang et al., 2019) and EPOET (Wang et al., 2020) are the founding algorithms of the field of *open-ended learning*, building upon prior approaches such as MCC (Brant & Stanley, 2017). This has led to an explosion of new use cases such as PINSKY (Dharna et al., 2020; 2022), which uses POET on 2D Atari games. This approach extends POET to generate 2D Atari video game levels alongside agents that solve these levels. Quessy & Richardson (2021) uses unsupervised skill discovery (Campos et al., 2020; Eysenbach et al., 2019; Sharma et al., 2020) in the context of POET to discover a large repertoire of useful skills. Meier & Mujika (2022) also investigate unsupervised skill discovery through reward functions learned by neural networks. Other uses of POET include the work by Zhou & Vanschoren (2022), who obtain diverse skills in a 3D locomotion task. POET has also been shown to aid in evolving robot morphologies (Stensby et al., 2021) and avoiding premature convergence which is often the result when using handcrafted curricula. Norstein et al. (2022) use MAP-Elites (Mouret & Clune, 2015) to open-endedly create a structu red repertoire of various terrains and virtual creatures. Hejna III et al. (2021) introduces TAME that evolves morphologies without tasks, potentially creating a system of open-ended morphology evolution.

Adversarial approaches are commonly adopted when developing open-ended algorithms. Dennis et al. (2020) propose PAIRED, a learning algorithm where an adversary would produce an environment based on the difference between the performance of an antagonist and a protagonist agent.

Domain randomization (Sadeghi & Levine, 2016), prioritized level replay (Jiang et al., 2021) and Adversarially Compounding Complexity by Editing Levels (ACCEL) (Parker-Holder et al., 2022) adopt a similar adversarial approach, where teacher agents produce environments and student agents solve them.

Several domains and benchmarks have been proposed with the aim of encouraging research into open-ended, general agents. Team et al. (2021) introduce the XLand environment, where a single agent is trained on $700k$ 3D games, including single and multi-agent games, resulting in zero-shot generalization on holdout test environments. Barthet et al. (2022) introduced an autoencoder (Kingma & Welling, 2013) and CPPN-NEAT based open-ended evolutionary algorithm to evolve Minecraft (Duncan, 2011; Cipollone et al., 2014) buildings. They showed how differences in the training of the autoencoders can affect the evolution and generated structures. Fan et al. (2022) create a Minecraft-based environment, MineDojo, which has numerous open-ended tasks. They also introduced MineCLIP as an effective language-conditioned reward function that plays the role of an automatic metric for generation tasks. Gan et al. (2021) introduce the Open-ended Physics Environment (OPEn) to test learning representations, and tested many RL-based agents. Their results indicate that agents that make use of unsupervised contrastive representation learning, and impact-driven learning for exploration, achieve the best result.

## 3 NEUROEVOLUTION OF AUGMENTING TOPOLOGIES

We leverage NeuroEvolution of Augmenting Topologies (NEAT) to evolve the structure of an agent's controller. NEAT starts with a population of simple neural networks (NNs), where the input neurons are directly connected to the output neurons without any hidden layers. Crossover is performed between two parents and the resulting children are mutated by adding connections and nodes, or perturbing weights. In this way, the NN will gradually be complexified. Crossover and mutations are illustrated in Section E.2 of Appendix E. One of the major problem to overcome is the *Permutations* or *Competing Convention Problem* Radcliffe (1993); Montana et al. (1989). Competing conventions describes the case in which the crossover of networks that represent the same solution but are encoded differently (e.g. a different ordering of neurons) can lead to a loss of information and a significantly worse child. NEAT addresses this by introducing a method to keep track of the historic origin of a gene by using the *innovation number*. Using this innovation number, identical genes from two parents can be aligned, while genes that only occur in one (denoted excess or disjoint genes depending on whether it occurs within or outside the range of the other parent's innovation numbers) can be inherited from the fitter parent. Finally, NEAT introduces speciation (Mahfoud, 1995), where individuals with similar topologies are grouped together, and share a fitness. This protects innovation and ensures diversity. This speciation calculation is shown by Equation 1 in Appendix D. In this equation, $c_1, c_2$, and $c_3$ are coefficients that indicate the importance of each factor while $N$ is the number of genes in the larger genome. $E$ and $D$ denote the number of excess and disjoint genes respectively. $W$ is the average weight difference of similar genes. $\delta$, then, indicates how close two genomes are; if $\delta$ is less than some threshold, then the two genomes belong to the same species.

NEAT has demonstrated superior performance when compared to fixed topology approaches, and has been used in numerous subsequent research works to great success (Stanley, 2007; Lehman et al., 2008; Stanley et al., 2009; Schrum et al., 2020; Clei & Bellec, 2022).

## 4 ENHANCED POET

Since our method is heavily based on EPOET, we briefly describe this method, as well as the original POET algorithm. **POET** focuses on evolving pairs of agents and environments in an attempt to create specialist agents that solve particular environments. POET uses modified version of `2D Bipedal Walker Hardcore` environment from OpenAI Gym (Brockman et al., 2016) as a benchmark. The first environment is a flat surface, and as evolution progresses, the environments become harder with the addition of more obstacles. POET also transfers agents across environments, which can prevent stagnation and leverage experience gained on one environment as a step towards solving another. An Environment-Agent (EA) pair is eligible to reproduce when the agent crosses a preset reward threshold on this environment. The next generation of environments is formed by mu-

tating the current population and selecting only those environments that are neither too easy nor too hard. Finally, environments are ranked by novelty, and only the most novel children pass through to the next generation. More information about the hyperparameters of POET is listed in Appendix C.

**EPOET** improves upon POET by adding in two algorithmic improvements: (1) a general method of evaluating the novelty of challenges and (2) an improved approach to deciding when agents should transfer to new environments. In the original POET, the way to evaluate novelty was to compare the *environment characterization* (EC) of different environments. This is obtained by using some fixed, domain-specific static features, such as the roughness of the terrain. This inherently limits the exploration of the algorithm, as it is restricted to explore within these preset confines. Enhanced POET introduces an improved EC, Performance of All Transferred Agents EC (PATA-EC), which is based on the performance of different agents in the environment. Secondly, the original transfer mechanism in POET was generally inefficient, as it increased the required computation (as each agent needed to be fine-tuned), and resulted in subpar transfers as it was too easy to qualify for transfer. Enhanced POET makes this process more strict, only transferring very promising agents.

Enhanced POET also improves upon the environmental encoding used in the original algorithm, which was fixed and thus had a limited number of unique and diverse environments it could represent. The solution to this problem is to use a more expressive encoding in the form of compositional pattern producing networks (CPPNs) (Stanley, 2007). A CPPN is a specific neural network, which can take in $x, y$ coordinates and produce a specific pattern when evaluated across an entire region. These CPPNs are evolved using the NEAT (Stanley & Miikkulainen, 2002) algorithm, which increases the complexity of the environments as evolution progresses.

Lastly, the authors introduce *Accumulated Number of Novel Environments Created and Solved* (AN-NECS), a metric for open-ended learning that, intuitively, describes the amount of new content that is generated by the algorithm. ANNECS counts the number of environments that satisfy two constraints: (1) it must neither be too easy nor too hard and (2) it must be eventually solved by some agents in the future. Thus, if the ANNECS metric increases as time goes on, it indicates that the algorithm is continually producing novel and meaningful environments.

## 5   OPEN-ENDEDLY EVOLVING THE TOPOLOGY OF AGENTS

Many of the approaches introduced in prior work have been implemented using a fixed topology approach in conjunction with optimizers such as evolutionary strategies (ES) (Salimans et al., 2017), V-MPO (Song et al., 2019) and Proximal Policy Optimization (Schulman et al., 2017), which motivates us to explore NEAT and the benefits it brings to the open-ended learning framework. We will describe the overall approach in Section 5.1.

### 5.1   AUGMENTATIVE TOPOLOGY ENHANCED POET (ATEP)

In this section, we discuss the basic building blocks of our algorithm and the different variants we experimented with. ATEP combines EPOET with NEAT to allow the agents' network topologies to evolve. This means that the algorithmic steps are very similar to EPOET, and the main differences are (1) the optimizer used: we use NEAT to optimize the variable-topology agents whereas EPOET used Evolution Strategies to optimize fixed-topology agents; and (2) the transfer mechanism, which will be discussed later in this section. The detailed flow of ATEP is described in Figure 1.

We first use NEAT to evolve a population for each environment. The valid environments (those that pass the minimal criterion) then reproduce to create a new generation of (slightly harder) environments. We then take the environment that is the most novel (as measured by the Euclidean distance between the *PATA-EC* scores), and create a new environment-agent pair. The transfer eligibility of these environments is then evaluated, and if there are valid transfers available, we can move agents between environments. In EPOET, transfer is performed as follows: we compare the fitness of the candidate agent to the fitness of the target agent, over the previous 5 generations. If the candidate's fitness is greater than all previous 5 fitness scores, we fine-tune it on the target environment and again compare it against the best fitness from the previous 5 generations. If both of these checks are passed we transfer the candidate and replace the target. For ATEP, we experiment with two different transfer mechanisms, the first being inspired by the approach used by EPOET, denoted as *Fitness-Based Transfer ATEP* (FBT-ATEP). In this case, we compare the best genome in the candi-

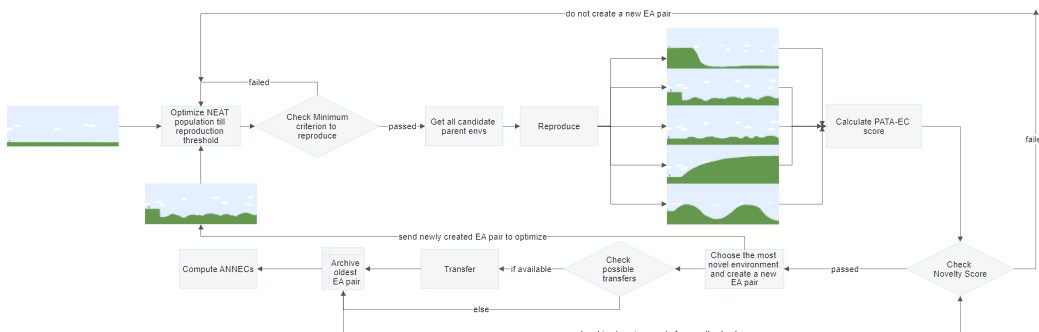

Figure 1: A flowchart demonstrating the flow of the ATEP framework, with blocks in green being where ATEP differs from POET. For both EPOET and ATEP, each environment is associated with an agent, represented by an ES population for EPOET and a NEAT population for ATEP. Mechanisms used in NEAT are descrtibed in Section 3 while the hyperparameters are in Table 3 of Appendix C. PATA-EC and ANNECS are described in Section 4 with other components used in EPOET. Section 5.1 describes the transfer mechanisms and Appendix D illustrates pseudocodes for the transfer mechanisms used in ATEP. The environment images used in the chart were created by ATEP.

date population to the best genome from the target population. We then perform the same checks as EPOET, and if both are passed, we replace the entire target population with the candidate.

For the second transfer mechanism, we use the speciation inherent in NEAT to influence transfer. Specifically, we check if the best genome in the candidate population is within a $\delta$ threshold (using the speciation calculation in Equation 1) of any target environment's best genome. If this is the case, we transfer the candidate species and replace the target species with it. This approach, called *Species-Based Transfer ATEP* (SBT-ATEP), skips the step of comparing fitness scores and has its own advantages which we discuss in the next section. Finally, we also consider random transfer (RT-ATEP) and no transfer (NT-ATEP) to investigate whether the transfer mechanisms have a large impact on the results.

## 5.2 EXPERIMENTAL SETUP

Now we describe the experimental setup for ATEP, its variants, and our baselines. In ATEP, we use NEAT as the algorithm to evolve the topology and weights.[1] To reduce the computational load, we change one aspect of the original EPOET paper, reducing the number of active environment-agent pairs from 40 to 20. We make this change for both EPOET and ATEP, so the results are still comparable.

We set up two baselines: the first, denoted as EPOET40x40, is EPOET with the original controller consisting of two hidden layers with 40 nodes each. The second baseline, EPOET20x20, is a controller with two layers of 20 nodes each. Having a baseline that has lesser nodes than the original EPOET's controller allows us to gather insights having lesser number of nodes, which we can then compare with larger number of nodes, which is the original EPOET's controller. This also allows us to evaluate the effect of having a small fixed topology, a comparatively larger fixed topology, and a variable topology. Furthermore, this allows us to confirm our hypothesis that fixed topology agents will stagnate after a certain level of complexity. For further details on the controllers, please refer to Appendix C.

## 6 RESULTS AND DISCUSSION

In this section, we discuss and analyze our results. We break the results into 3 different categories: Open-Endedness, nodes complexity exploration and generalization ability. All results are gathered based on 4 seeds due to the expensive computational load, with each algorithm requiring approxi-

---

[1]Hyperparameter settings for the various methods are listed in Appendix C.

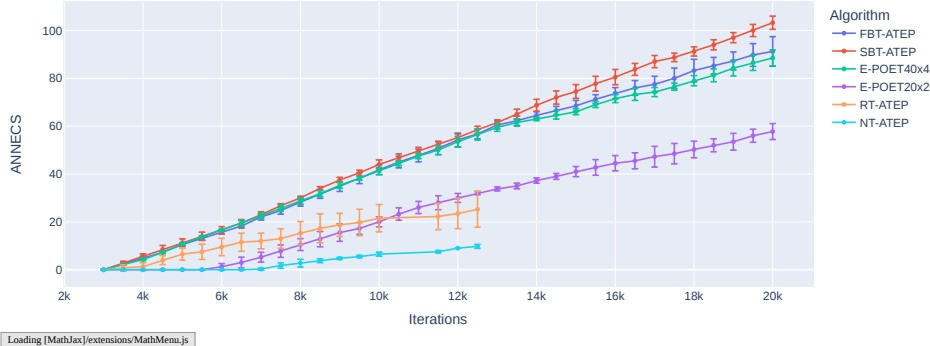

Figure 2: Accumulated Number of Novel Environments Created and Solved (ANNECS). The results are gathered on 4 seed. Solid line represents mean and the error bars represent standard daviation across the seeds. To save compute, we stop the NT- and RT-ATEP experiments early, as it is clear that they perform poorly.

mately $50,000$ to $200,000$ CPU hours for a single run. EPOET20x20 required the least amount of computation, while SBT-ATEP required the most. Each algorithm was run in parallel on a cluster consisting of 264 Intel Xeon cores, with the runtime ranging between 10 and 30 days. Each algorithm had a maximum of a $10^{12}$ function evaluations per run. One function evaluation means one run of an episode that could be 2000 timesteps long.

The change from 20 to 40 active environment-agent (EA) pairs limits the agent to have minimum number of optimization steps to 3000 from 6000, which was the case in the original EPOET research. This number is calculated as we try to make an EA pair every 150 iterations and we have 20 active EA pairs, thus it becomes possible that if an EA pair is created every 150 iterations and active EA pair threshold is reached, we archive the oldest. Thus, giving less time for the agent to optimise.

## 6.1 OPEN-ENDEDNESS

As mentioned, Wang et al. (2020) introduce the ANNECS metric to capture the open-endedness of an algorithm; we take it as our most important score to judge which algorithm performs better on complex environments. Refer to Appendix F for more explanation on ANNECS.

Figure 2 shows the ANNECS score as a function of training time. We see that there is a significant difference between EPOET20x20 and FBT- and SBT-ATEP, indicating that the small network results in solving fewer environments. EPOET40x40 performs substantially better than EPOET20x20, and is competitive with ATEP early on during training. The rate of increase in ANNECS, however, does decrease after about 13k iterations, whereas ATEP increases at a consistent rate. This substantiates our hypothesis that fixed topology agents will start stagnating at some level of environment complexity, due to capacity issues. While we can improve the results by increasing the size of the network, that will merely delay the onset of performance plateau.

FBT-ATEP outperforms EPOET40x40, although it also slows down slightly as time progresses. This is due to replacing the entire target population with the transferred population, which may eliminate all useful skills learned by the target population. SBT-ATEP, on the other hand, only replaces a single species that is close to the candidate species, leaving the rest of the population intact. We also find that SBT-ATEP has negligible performance plateaus during the run of our experiments in solving environments and, even though it performed similarly to FBT-ATEP and EPOET40x40 early on during training, it starts to outperform these in the second half of the experiment. This, as we will show later, is partly due to SBT-ATEP exploring more actions. We further note that the variations using no transfer (NT-ATEP) or random transfer (RT-ATEP) perform poorly, indicating that intelligent transfer mechanisms are necessary.

Although ATEP outperforms EPOET, it is more computationally expensive, as measured by the number of function evaluations. One function evaluation means one individual being evaluated on

an environment. SBT-ATEP has the most function evaluations since once a species transfers from one population to another, it becomes highly probable that it can transfer in the opposite direction because they may now be within the $\delta_{threshold}$ range. This increases the population size, resulting in more function evaluations. The tradeoff here is of function evaluations to performance, which is justified as the performance confirms our hypothesis. Figure 3 displays the total number of function evaluations.

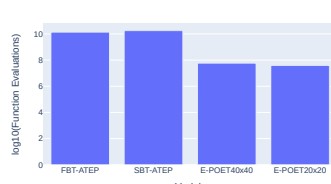

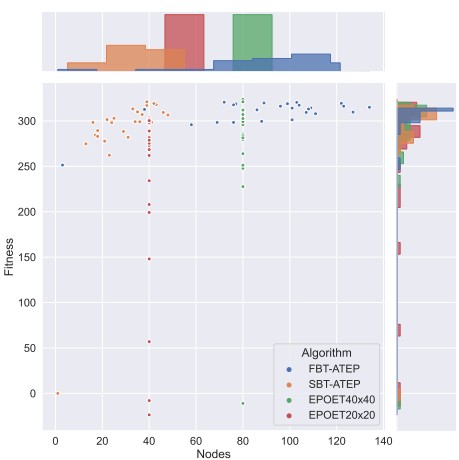

Figure 3: Cumulative sum of the number of function evaluations, with the Y-axis converted to a log-scale. While ATEP requires significantly more function evaluations, we find that its total wall-time is only 3 times more than EPOET, as the neural networks are generally smaller and each evaluation does not take as long.

Figure 4: Mapping of fitness to nodes. We plot values every 1000 iterations, starting at iteration number 150. Each dot represents a specific iteration, as well as the mean fitness over all environments and the mean number of nodes in the population. Distribution on top represents distribution of nodes while distribution on the left represents fitness distribution.

## 6.2 NODES COMPLEXITY EXPLORATION

We have now shown that SBT-ATEP outperforms all of the other tested methods based on the AN-NECS score. We also find that it generally uses a smaller neural network with fewer nodes than the other algorithms. Figure 4 shows the number of nodes and corresponding fitness value for each algorithm. We can see that SBT-ATEP generally has a high fitness, but fewer nodes than the other approaches. This is echoed in Figure 5c, where SBT-ATEP has the least number of nodes for most of the experiment, although it gradually adds nodes and complexity. FBT-ATEP, on the other hand, adds nodes very rapidly. This again indicates that the transfer mechanism in EPOET is critical. Refer to Appendix E for evolved networks from SBT-ATEP to illustrate emerging network complexity.

Inspired by this, we further look into a simple *Fitness to Nodes* ratio (FNR) metric, shown in Figure 5a, and find that SBT-ATEP outperforms all other algorithms on this metric for the majority of the run. This indicates that SBT-ATEP outperforms all other algorithms on a per-environment basis, while using fewer nodes. This leads us to believe that a better-curated transfer mechanism, based on SBT-ATEP, will sustain the FNR for longer runs.

Furthermore, in Figure 5b, we calculate an *ANNECS to Nodes* ratio (ANR) metric with the intent to observe the role of nodes in the Open-Endedness of the agents, i.e. to have the ability to complexify over time. We observe that SBT-ATEP performs significantly better than the other models. FBT-ATEP has the lowest ANR, as it adds nodes much faster than the rate of increase in ANNECS.

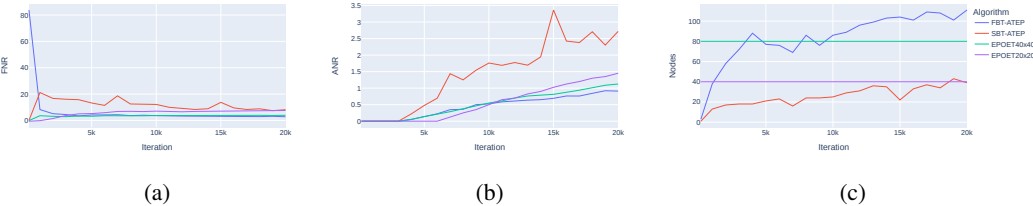

Figure 5: Analysis with respect to the number of nodes. Figure **(a)** shows FNR along iterations, **(b)** shows ANR along iterations, **(c)** shows the addition of nodes along iterations

### 6.3 GENERALIZATION ABILITY

We next evaluate the generalization ability of our open-ended agents, as prior work (Team et al., 2021) has shown that these agents have the potential to generalize to new unseen environments. To concretely test this, we first take the 20 latest environments from each method. For each environment, we take the latest agent that could solve this environment from the method under consideration. Each of these agents is now evaluated on the selected environments from the other methods (60 in total). We perform 30 runs per environment-agent pair and calculate the mean and maximum of the rewards. We split the results into three categories: environments with fitness scores above 300, between 200 and 300, and below 200. Scores below 200 indicate that the environment has not been solved by the agent. Figure 6d shows the performance of each method when evaluated on the 60 other environments. We observe that SBT-ATEP outperforms all other models, with only $10\%$ of the environments remaining unsolved.

Secondly, we test the generalization capabilities of agents on all of the environments created by their own algorithm. We exclude EPOET20x20 as it fails to solve 80 environments in the whole run. We take into account 80 environments that were solved by the model itself and observe how each agent performs on all of them. Figures 6a, 6b and 6c show the results. Here, early-stage agents perform worse and late-stage agents are shown to have generalization abilities on previously unseen landscapes. The transfer mechanism plays a key role in this generalization, as it exposes agents to more environments. Despite not having seen all environments, late-stage agents generalize much better. SBT-ATEP generalizes the best, with the lowest proportion of unsolved environments, in contrast to the lower-performing EPOET40x40 and FBT-ATEP.

Finally, we briefly investigate potential reasons why SBT-ATEP outperforms FBT-ATEP. We find that SBT-ATEP explores more actions, as it only transfers a single species instead of replacing the target population as is done by FBT-ATEP. This allows the new species to complement the actions that were already explored by the existing population. Appendix B shows the action distribution of each action for SBT-ATEP, FBT-ATEP and EPOET40x40.

## 7 CONCLUSION AND FUTURE WORK

This work investigated the effect of having an Augmentative topology agent on an open-ended learning algorithm's performance. We hypothesized that using a fixed topology would result in agents that exhibit delays in solving an environment after a certain point in environment complexity. We showed that this is indeed the case, and addressed this limitation by introducing ATEP, which allows the network topology of the agents to change and add complexity as necessary. We demonstrated that this approach outperforms existing methods in terms of the ANNECS score and generalization ability, while using fewer nodes in the neural networks. Our approach, however, does require more function evaluations than competing approaches. Thus, a promising future direction would be to use NEAT with Novelty Search (Lehman et al., 2008) or Surprise Search which tends to converge faster than simple NEAT (Gravina et al., 2016). QD algorithms may also be worthwhile to explore in the context of open-ended learning as they have the ability to produce a population of high-performing and diverse individuals (Bhatt et al., 2022). Exploring Neurogenesis (Maile et al., 2022; Draelos et al., 2017), where neurons are added to a single neural network based on various external triggers, could also be a promising direction. To reduce computational load, it would also be promising to

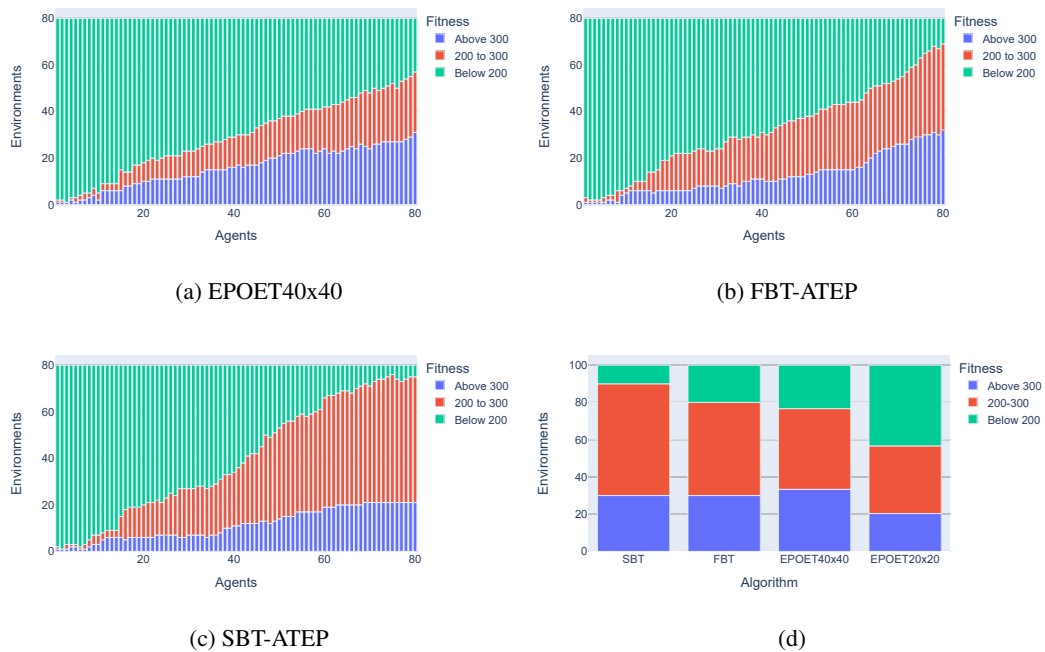

Figure 6: Figures showing generalization capabilities. Figures **(a)**, **(b)** and **(c)** show agents of 80 solved environments being tested on all 80 environments, for EPOET40x40, FBT-ATEP and SBT-ATEP respectively. Note that EPOET20x20 does not take part in this test as it failed to produce 80 environments in the run. Figure **(d)** shows each algorithm being tested on the 20 latest environments created by all other algorithms, i.e., each algorithm is evaluated on 60 environments. The Y-axis shows the percentage of environments in each category. Each test is conducted for 30 runs and the mean scores are taken.

look into developing single-population open-ended learning methods without losing the exploration abilities of EPOET.

Furthermore, we have opened up possible future research into transfer mechanisms. We compared simple approaches such as FBT and SBT, but more advanced approaches could yield further performance improvements. For instance, we could combine both FBT and SBT in a weighted manner, or transfer only a certain percentage of a species or population. Finally, this work provides a starting point, like EPOET itself, into open-ended learning with augmentative topology agents. We therefore used the modified version of 2D BipedalWalker as our benchmark. Future work should compare ATEP with standard EPOET on different and more complex environments. Ultimately, we hope that this new approach furthers research into open-ended algorithms that do not slow down over time, and can keep up with an ever-changing environment.

## REPRODUCIBILITY AND ETHICAL STATEMENT

For reproducibility, we have provided a GitHub repo[2] where users can follow instructions to reproduce the experiments. We have also provided detailed hyperparameter tuning in Appendix C and pseudocodes in Appendix D for user reference. ATEP is an Open-Ended Learning algorithm that has stochastic elements, similar to many other machine learning algorithms. It is critical for users to perform standard evaluations as the user would do for other machine learning algorithms. A full run of ATEP may be computationally expensive and will take approximately $50,000$ to $200,000$ CPU hours.

---

[2]The GitHub repo will be provided after acceptance, but a zip file containing the source code will be submitted alongside this paper.

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

## A ABBREVIATIONS

Table 1: Abbreviations for commonly used terminologies.

| Abbreviation | Termonology |
| --- | --- |
| MCC | Minimal Criterion Coevolution |
| POET | Paired Open-Ended Trailblazer |
| EPOET | Enhanced Paired Open-Ended Trailblazer |
| NEAT | NeuroEvolution of Augmenting Topologies |
| ATEP | Augmentative Topology Enhanced POET |
| NN | Neural Network |
| FBT-ATEP | Fitness-Based Transfer ATEP |
| SBT-ATEP | Species-Based Transfer ATEP |
| RT-ATEP | Random Transfer ATEP |
| NT-ATEP | No Transfer ATEP |
| EPOET40x40 | EPOET with NN of two hidden layers, 40 nodes each |
| EPOET20x20 | EPOET with NN of two hidden layers, 20 nodes each |
| ANNECS | All New and Novel Environments Created and Solved |
| PATA-EC | Performance of All Transferred Agents - Environment Charactization |
| EA | Environment-Agent |
| FNR | Fitness to Nodes Ratio |
| ANR | ANNECS to Nodes Ratio |

## B ACTION DISTRIBUTION FIGURES

In Figure 7 we have added figures for the action distributions of each algorithm.

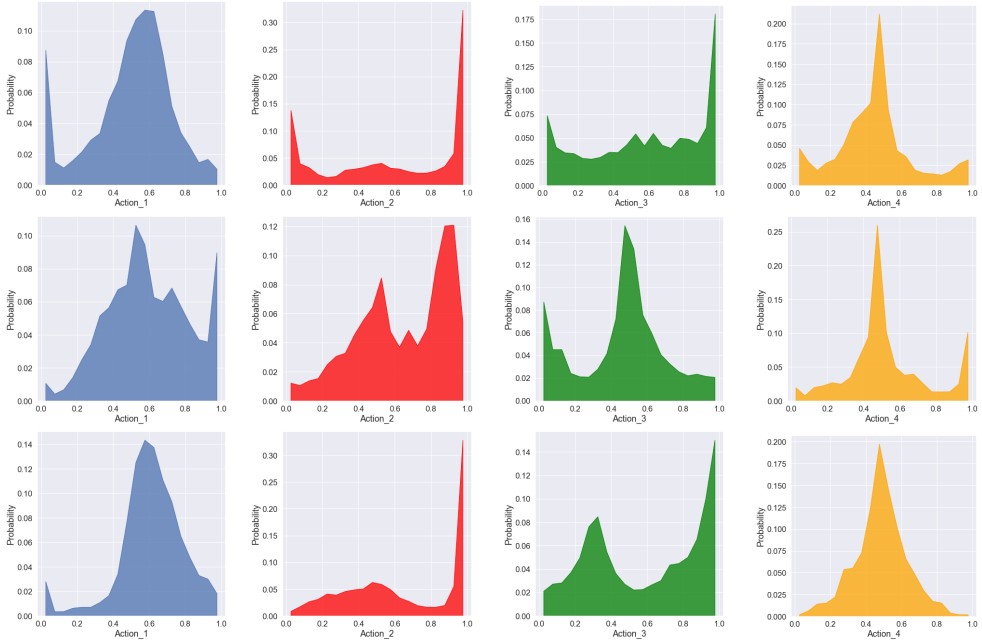

Figure 7: Action Distributions for (top row) SBT-ATEP, (middle row) FBT-ATEP and (bottom row) EPOET40x40. Each column represents one specific dimension of the action array.

## C  HYPERPARAMETER SETTINGS

This appendix is for hyperparameter settings which were used for ATEP and the baselines. Table 2 shows hyperparameters for ES in EPOET. Table 3 shows settings for NEAT in ATEP and Table 4 shows parameter configurations for CPPNs. General hyperparameters for reproduction are given in Table 5.

Table 2: ES hyperparameter settings

| Hyperparameter | Setting |
| --- | --- |
| ES Population | 512 |
| Weight update method | Adam |
| Initial learning rate | 0.01 |
| Decay factor of learning rate | 0.9999 |
| Initial noise standard deviation | 0.1 |
| Lower bound of noise standard deviation | 0.01 |
| Decay factor of noise standard deviation | 0.999 |

Table 3: NEAT hyperparameter settings

| Hyperparameter | Setting |
|---|---|
| Population size | 1000 |
| Crossover probability | 0.3 |
| Weight mutation (small) probability | 0.85 |
| Weight mutation (large) probability | 0.15 |
| Weight mutation (small) range | -0.1 - +0.1 |
| Weight mutation (large) range | -1 - +1 |
| Connection mutation probability | 0.85 |
| Node mutation probablity | 0.15 |
| Maximum stagnation | 60 |
| $c_{1}$ | 1.0 |
| $c_{2}$ | 1.0 |
| $c_{3}$ | 3.7 |
| Delta threshold | 3.0 |
| Initial condition | full |
| Activation function | tanh |
| Number of inputs | 24 |
| Number of outputs | 4 |

Table 4: CPPN hyperparameter settings

| Hyperparamer | Setting |
|---|---|
| Initial condition | full |
| Activation default | identity |
| Activation options | identity sin sigmoid square tanh |
| Aggregation default | sum |
| bias init stdev | 0.1 |
| bias init type | gaussian |
| bias max value | 10.0 |
| bias min value | -10.0 |
| bias mutate power | 0.1 |
| bias mutate rate | 0.75 |
| num inputs | 1 |
| num outputs | 1 |
| response init mean | 1.0 |
| response init type | gaussian |
| response max value | 10.0 |
| response min value | -10.0 |
| single structural mutation | True |
| structural mutation surer | default |
| weight init stdev | 0.25 |
| weight init type | gaussian |
| weight max value | 10.0 |
| weight min value | -10.0 |
| weight mutate power | 0.1 |
| weight mutate rate | 0.75 |

Table 5: EPOET general hyperparameter settings for ATEP

| Hyperparameter | Setting |
|---|---|
| Reward threshold | 200 |
| Environment difficulty MC | 25 - 340 |
| Transfer check | 25 |
| Reproducibility check | 150 |
| Active environments | 20 |

## D TRANSFER MECHANISMS

This appendix shows pseudocodes of Species-based and Fitness-based transfer mechanisms. Algorithms are shown in Algorithm 1 and 2, respectively. 2, in particular, is very similar to the transfer algorithm described by Wang et al. (2020). $find\_delta(.)$ in Algorithm 1 is calculated by the equation Equation 1.

$$\delta = \frac{c_1 E}{N} + \frac{c_2 D}{N} + c_3 \cdot W \tag{1}$$

---

**Algorithm 1:** Species-Based Transfer

---

**Input** : Candidate population's best individual $I_c$. A function $find\_delta(.)$ that calculates
delta score and $\delta_{threshold}$.

Let $M$ = All environments - {Candidate environment}

**foreach** $m \in M$ **do**

    $I_m$ = best individual of environment $m$

    $\delta_{ct}$ = find_delta($I_c$, $I_m$) using Equation 1

    **if** $\delta_{ct} \leq \delta_{threshold}$ **then**

        delete target species

        Transfer candidate species to target population

    **else**

        Transfer is not possible

    **end**

**end**

---

**Algorithm 2:** Fitness-Based Transfer

---

**Input** : Candidate population's best individual $I$, a function *Score(.)* that calculates the
maximum of the target agent's 5 most recent fitness scores.

Let $M$ = All environments - {Candidate environment}

**foreach** *m in M* **do**

    Compute direct transfer $I_D$;

    **if** $I_D > Score(m)$ **then**

        Compute fine-tuning transfer $I_P$;

        **if** $I_P > Score(m)$ **then**

            Add m to $T\_candidates$

        **else**

            Transfer not possible

        **end**

    **else**

        Transfer not possible

    **end**

**end**

Delete whole population of $T\_candidates$

Transfer whole candidate population to $T\_candidates$

---

# E EVOLVED NETWORKS

## E.1 TOY EXAMPLE

The toy example is presented to illustrate mutation and crossover operators in NEAT.

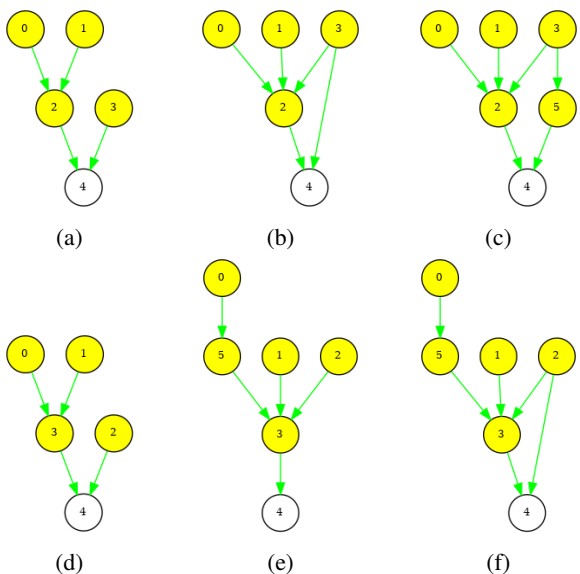

Figure 8: A toy example of the mutation of an individual in NEAT. **(a)** shows a simple network, **(b)** mutates by adding a connection from node 3 to node 2 and **(c)** mutates by adding node 5 between node 3 and node 2. **(d)**, **(e)** and **(f)** shows a toy example of the crossover between two individuals of NEAT. **(d)** and **(e)** shows two individuals. **(f)** is the result of the crossover between **(d)** and **(e)**, which results in adding a new connection from node 2 to node 4 in individual **(e)**. These illustrations are motivated by NEATs' original paper (Stanley & Miikkulainen, 2002).

## E.2 AUGMENTED NEURAL NETWORKS

For illustration purposes we took a later stage agents' neural network (NN) which eventually became the best performing individual for the solved environment. Yellow nodes represent inputs, aqua represents hidden nodes while white maps to outputs. Figure 9a shows the start of the NN without any evolution applied. Figure 9b is the state after 2000 iterations. It shows to have a few nodes in hidden layer one and a couple of nodes in hidden layer two. Complexity has not yet been emerged as it can be observed. Figure 10a is the NN after 4000 iterations. We can observe emerging complexity. Some residual connections can be seen, for example node 5, an input node connects to hidden node 30, but also connects to an output node 26. Figure 10b is the evolved network that became the best individual in the population, after 6000 iterations of when it was created. Here complexity is observed in many levels. Firstly, an input node 2 connects to input node 0, where as node 46 appears to be having a forward pass to the input 2 and 0. Secondly, we observe an interesting small block of 6 nodes conventionally fully connected, input 22 and 23 connected to 40 and 42 which further connects to 41 and 43, that connects to the outputs. Overall depth of the network seems to be variable.

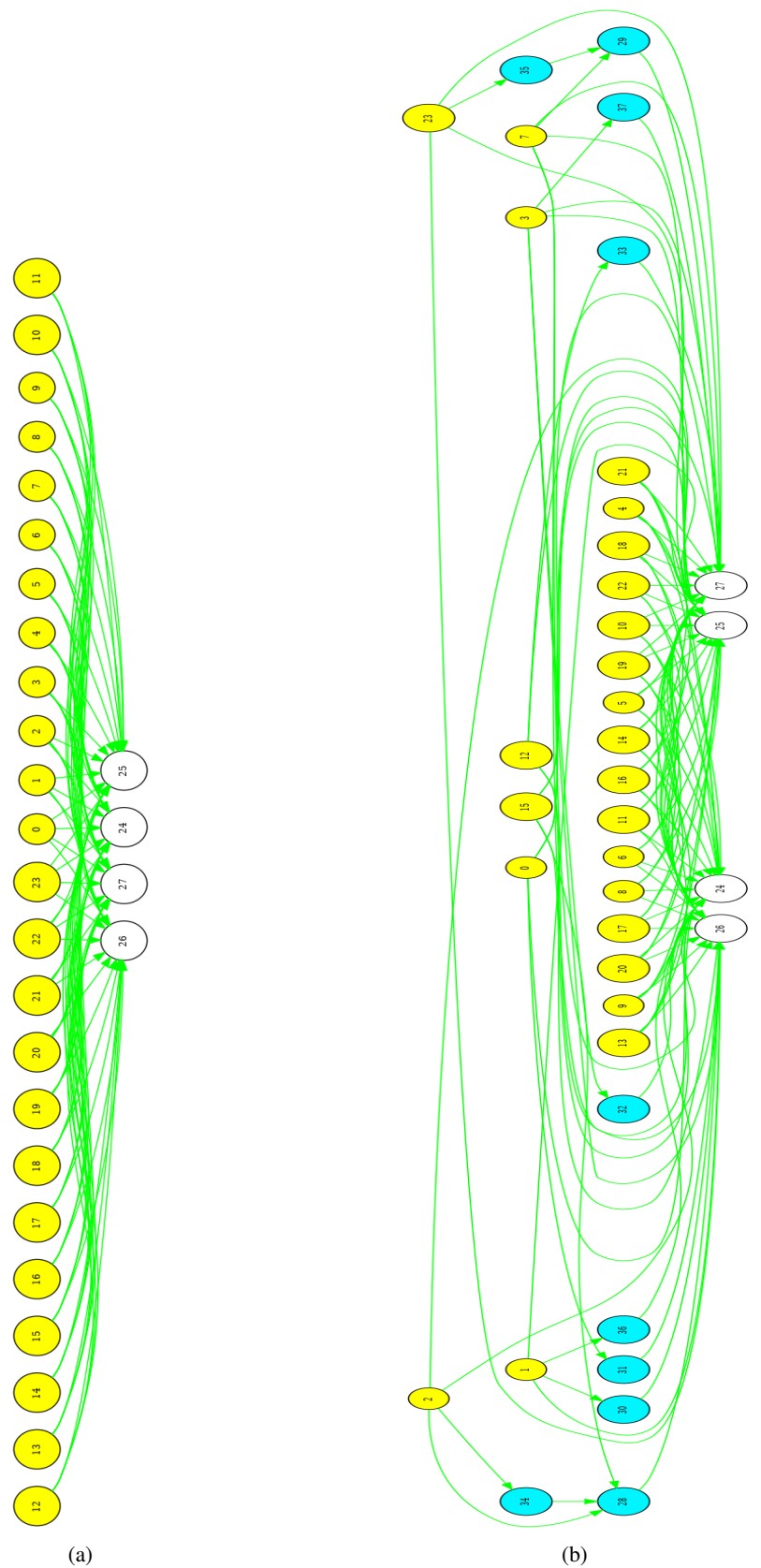

Figure 9: From left to right: NNs at iteration 0 and 2000 of when it was created.

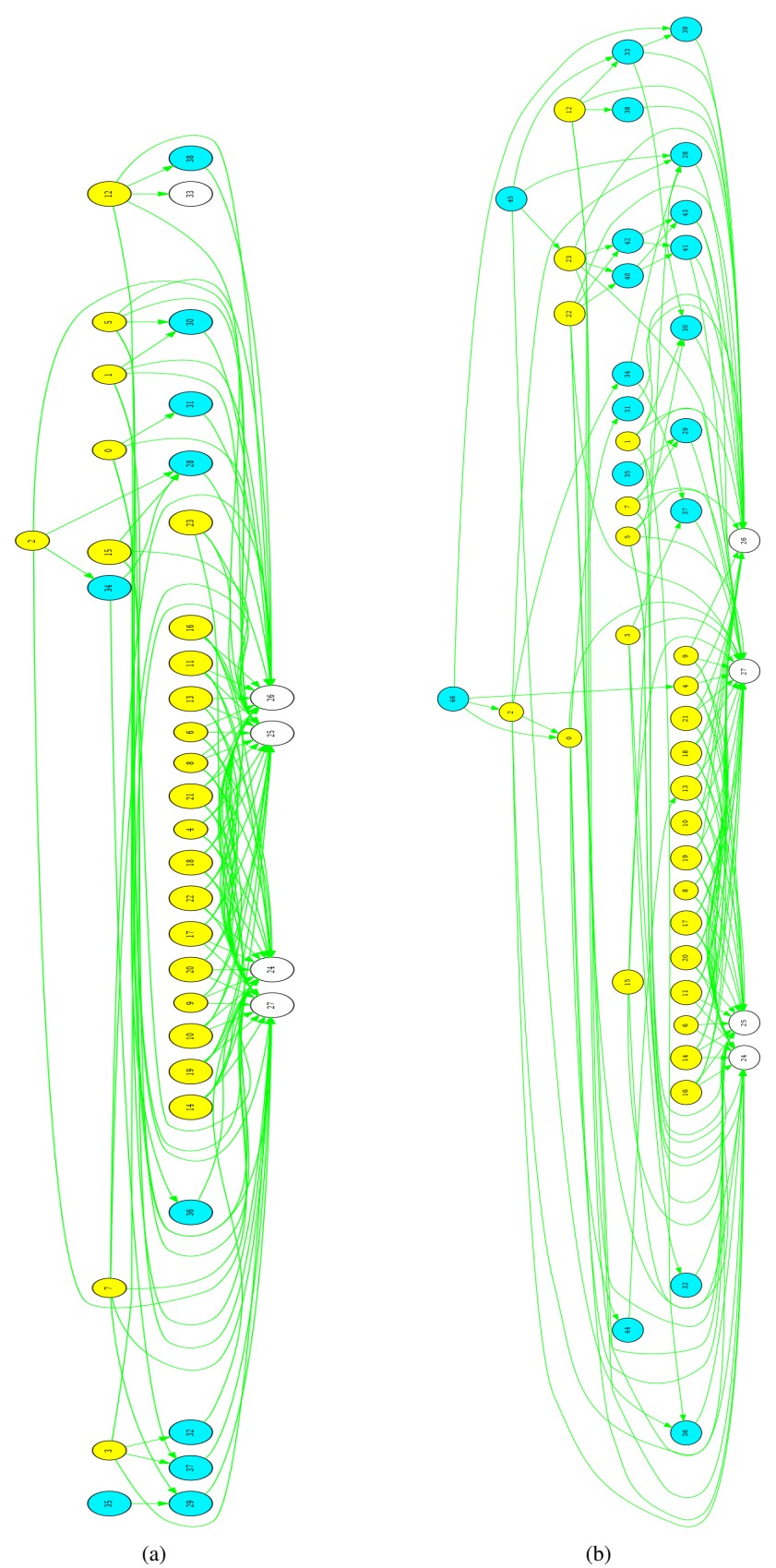

Figure 10: From left to right: NNs at iteration 4000 and 6000 of when it was created.

# F    KEY CONCEPTS FROM EPOET

## F.1    PATA-EC

*Performance of All Transferred Agents-Environment Characterization* (PATA-EC) is introduced by EPOET as a domain-general environment characterization score. PATA-EC is a score that describes the environment behaviour through all active and archived agents behaviour on the environment. Four steps to calculate PATA-EC are: (1) **Evaluate**: Each created environment evaluates all agents and stores their raw scores in a vector. (2) **Clip**: The cores are clipped between a lower and upper bound. By doing so, we clip the possibilities of having an agent that is a failure (too low score) or the environment is to easy for the agent (too high score). (3) **Rank-normalize**: We the sort the raw scores by their rankings, and normalize the scores in between the range of [-0.5,0.5]. This allows us to use direct euclidean distance on the PATA-EC scores. Eventually PATA-EC score helps in creating most novel environments and is not domain-specific thus could be used in any environment.

## F.2    ANNECS

*All New and Novel Environments Created and Solved* is a measure of progress in an open-ended system. The intuition behid it is to see if the system is generating environments that are novel but also eventually gets solved by the agents, thus measuring useful environments. The environments that are useful gets solved by an agent thus progressing itself in an open-ended system. To be counted in ANNECS score, an environment created at a particular iteration must: **(1)** pass the minimal criteria (i.e. not too easy or too hard) measured against all active and archived agents generated over the entire run so far, **(2)** eventually solved by the system, which means the algorithm will not receive credit for producing unsolvable environments. As ANNECS metrics constantly goes up indicates the algorithm is making meaniungful environments.

