# OpenReview forum: "Augmentative Topology Agents For Open-Ended Learning"
_ICLR.cc/2023/Conference — Submitted to ICLR 2023_

### Official Review · Reviewer_TYB9 · 2022-10-23

**Confidence:** 4
**Correctness:** 3
**Technical Novelty And Significance:** 2
**Empirical Novelty And Significance:** 2
**Recommendation:** 3

**Clarity, Quality, Novelty And Reproducibility:**

The paper is well-written overall. However, several key concepts were used in the paper without thorough definition.

The paper introduces a seemingly straightforward adoption of NEAT in an existing framework for open-ended learning. The novelty of the paper does not seem to be strong. The technical novelty and contribution of the newly developed transfer mechanisms also require more justifications.

**Strength And Weaknesses:**

Strength:It is interesting and important to study the problem of open-ended learning. It is also interesting to consider structure/topology evolution in order to improve the effectiveness of evolved NNs towards solving complex problem instances.

Weakness:This paper is motivated by the assumption that generalization can be improved by allowing evolved NNs to become increasingly complex. However, the validity of this assumption may be questionable. In fact, as the experiment results showed, the NNs evolved by NEAT tend to have small sizes, which help to maintain good generalization performance. Hence the experiment results reported do not seem to be fully compatible with the key assumption of the algorithm design.

Several key concepts were used in the paper without thorough definition. For example, it remains unclear what "completely open-ended" on page 1 means. How to measure the degree of open-endedness? In line with this question, it remains unclear how open-ended learning is defined. What does it mean by "complexity arises randomly"? I cannot find any thorough mathematical definition of this learning problem. It is also unclear how the open-ended learning problem is related to other learning paradigms such as transfer learning and multitask learning.

Furthermore, the practical significance of open-ended learning is not investigated with sufficient depth in the paper. In Section 2, the authors introduced some practical applications of open-ended learning. However, it seems that none of such applications were investigated in the paper based on the newly proposed algorithm. Hence, the practical usefulness of the new algorithm remains questionable to a certain extent.

One major contribution of this paper appears to be using NEAT instead of ES (or other reinforcement learning algorithms) to drive the evolution of NNs. The introduction of NEAT in Subsection 4.1 appears to be following exactly the original design of NEAT. It is not clear what are the technical difficulties of using NEAT for open-ended learning and how the technical difficulties were actually tackled in this paper. A direct adoption of NEAT does not seem to be sufficiently novel. Furthermore, since Subsection 4.1 introduces only an existing algorithm, it may not be appropriate to put it in the section on new methods (or new algorithms).

Another major concern is regarding the proposed transfer mechanisms. According to the paper, "transfers agents across environments can prevent stagnation and leverage experience gained on one environment as a step towards solving another". However, what kind of "stagnation" is considered in the paper? Is this a common problem for existing algorithms, such as POET and EPOET? Why can the newly introduced transfer mechanism prevent stagnation? Without clear answers to all these questions, the technical novelty and contribution of the newly developed transfer mechanisms remain doubtful. The mechanism design also appears to be heuristic in nature. It is not clear what theoretical principles are being followed while designing the transfer mechanisms.

More technical details should be provided in the paper. For example, the first criteria for environment counting on page 3 is that the environment must neither be too easy nor too hard. However, what are the precise definitions of easy and hard environments and why?

The new algorithm appears to have several new hyper-parameters to be fine-tuned, for example the threshold delta. It may not be easy to tune these hyper-parameters. Hence it may be practically difficult to use the new algorithm for some real-world open-ended learning applications.

The new algorithm is computationally costly to run (up to 200k cpu hours according to the paper). As a result, only 2 seeds have been tested in the paper. It may not be easy to draw any solid conclusions from the experiment results on 2 seeds. Furthermore, the network architectures (40x40 or 20x20) experimented appear to be much smaller than the typically used architectures among the majority of deep reinforcement learning algorithms. It is unclear whether the new algorithm can still outperform the baselines upon increasing the complexity of the network architectures. In general, as the benchmark problem studied is simple, the claim that NEAT enables the new algorithm to solve challenging or complex problems is not strongly verified in the paper.

**Summary Of The Paper:**

This paper proposes to use NEAT to evolve NNs with varying structures to solve an increasingly complex collection of reinforcement learning problems under the framework of open-ended learning. It also studied two alternative transfer mechanisms to transfer evolved NNs across different environments. Experiments show that the newly proposed algorithm may achieve good performance for one benchmark of open-ended learning.

**Summary Of The Review:**

It is interesting and important to study the problem of open-ended learning. However, the technical novelty of the paper does not seem to be sufficiently strong. The newly proposed algorithm is also computationally costly to run. Consequently, the practical usefulness of the new algorithm requires more investigations.

---

> ### Author Response · Authors · 2022-11-18
> **Thank you for the detailed review and addressing all comments**
>
> We would like to thank you reviewer TYB9 for the detailed reviews. It will highly improve the paper and hopefully benefit the research community. Following are reviews with comments on how we addressed them. If the space limit for comments end, we will add extension in the comment.
>
> > This paper is motivated by the assumption that generalization can be improved by allowing evolved NNs to become increasingly complex. However, the validity of this assumption may be questionable. In fact, as the experiment results showed, the NNs evolved by NEAT tend to have small sizes, which help to maintain good generalization performance. Hence the experiment results reported do not seem to be fully compatible with the key assumption of the algorithm design.
>
> To clarify, the key assumption is to have an augmentative topology agent that adds nodes and connections over time so that the structure complexifies and thus can handle complex environments, and not to have a larger size of the neural network through the augmentation. We further added some examples of evolving networks from our runs to illustrate the emerging complexity. It is added in Appendix E.
>
> > Several key concepts were used in the paper without thorough definition. For example, it remains unclear what "completely open-ended" on page 1 means. How to measure the degree of open-endedness? In line with this question, it remains unclear how open-ended learning is defined. What does it mean by "complexity arises randomly"? I cannot find any thorough mathematical definition of this learning problem. It is also unclear how the open-ended learning problem is related to other learning paradigms such as transfer learning and multitask learning.
>
> Thank you for notifying us. We have further added "where completely open-ended means to run indefinitely and create novel artifacts" in Section 1 to address our reviewer's comment.
>
> In our experiments, the degree of open-endedness is measured by the ANNECS score.
>
> To clarify, in the context of Minimal Criterion Coevolution (MCC) algorithm [1], randomly arising complexity means that the algorithm is not robust enough to efficiently create complex environments, because environments created should be solved by one of the agents and are thus very easy.
>
> > Furthermore, the practical significance of open-ended learning is not investigated with sufficient depth in the paper. In Section 2, the authors introduced some practical applications of open-ended learning. However, it seems that none of such applications were investigated in the paper based on the newly proposed algorithm. Hence, the practical usefulness of the new algorithm remains questionable to a certain extent.
>
> To clarify, the paper is written with an intention to focus on the benefits of augmentative topology agents in an open-ended learning paradigm, and not to demonstrate practical applications. We follow benchmarks from the original EPOET paper to demonstrate the effectiveness of our technique. We have added more motivation in the introduction section 1 by adding "Through the motivation by the fact that humans have always learnt and innovated in an open-ended manner, Open-ended learning research field emerged. For instance, humans did not invent microwaves to heat food, but to study radars. Vacuum tubes and electricity was invented for very different reason but we stumbled upon computers through them." [2]
>
> > One major contribution of this paper appears to be using NEAT instead of ES (or other reinforcement learning algorithms) to drive the evolution of NNs. The introduction of NEAT in Subsection 4.1 appears to be following exactly the original design of NEAT. It is not clear what are the technical difficulties of using NEAT for open-ended learning and how the technical difficulties were actually tackled in this paper. A direct adoption of NEAT does not seem to be sufficiently novel. Furthermore, since Subsection 4.1 introduces only an existing algorithm, it may not be appropriate to put it in the section on new methods (or new algorithms).
>
> The novelty of the paper lies in the effect of using augmentative topology agents. We mention in our future works which other algorithms we can use instead of NEAT. As NEAT (or any other algorithm that augments neural network topologies) has not been used in an Open-Ended Learning framework, we suggest it as a novel contribution as well. We have also rearranged the subsections as advised by the reviewer.

---

> > ### Author Response · Authors · 2022-11-18
> > **Continuation of the comments**
> >
> > > Another major concern is regarding the proposed transfer mechanisms. According to the paper, "transfers agents across environments can prevent stagnation and leverage experience gained on one environment as a step towards solving another". However, what kind of "stagnation" is considered in the paper? Is this a common problem for existing algorithms, such as POET and EPOET? Why can the newly introduced transfer mechanism prevent stagnation? Without clear answers to all these questions, the technical novelty and contribution of the newly developed transfer mechanisms remain doubtful. The mechanism design also appears to be heuristic in nature. It is not clear what theoretical principles are being followed while designing the transfer mechanisms.
> >
> > We consider stagnation in terms of performance plateau in the ANNECS score as this score measures the Open-Endedness of the algorithm. We have edited the paper as advised.
> >
> > Yes, this issue has been spotted in EPOET, where the authors accept that performance plateaus in terms of ANNECS score, which is associated with environment difficulty and capacity of agents to learn [3]
> >
> > We have explored the area of the reason why our transfer mechanism does not plateau during our experiment by having a look into the actions explored. We added these distributions in the Appendix B.
> >
> > We agree with the reviewer that the mechanism is heuristic in nature, as evolutionary algorithms are heuristic. We have tried to explain the theoretical principles of transfer mechanisms by providing the speciation formula in Appendix D and notified that it is motivated by NEAT.
> >
> > > More technical details should be provided in the paper. For example, the first criteria for environment counting on page 3 is that the environment must neither be too easy nor too hard. However, what are the precise definitions of easy and hard environments and why?
> >
> > We have used the PATA-EC score (which is introduced by EPOET) as the measure of being not too easy nor too hard. We have added more details in Appendix F, as advised by the reviewer.
> >
> > > The new algorithm appears to have several new hyper-parameters to be fine-tuned, for example the threshold delta. It may not be easy to tune these hyper-parameters. Hence it may be practically difficult to use the new algorithm for some real-world open-ended learning applications.
> >
> > We agree with the fact that NEAT brings in 7 more hyperparameters which we believe are needed as we have a full population of individuals to evolve. In our experiments we did very limited tuning and got good results which indicated that default NEAT hyperparamers are very useful and does not require too much tuning. Similarly, CPPN also has many hyperparameters which is due to the population of individuals. Thus we would say that this is the case with many evolutionary algorithms and reinforcement learning algorithms as well.
> >
> > > The new algorithm is computationally costly to run (up to 200k cpu hours according to the paper). As a result, only 2 seeds have been tested in the paper. It may not be easy to draw any solid conclusions from the experiment results on 2 seeds. Furthermore, the network architectures (40x40 or 20x20) experimented appear to be much smaller than the typically used architectures among the majority of deep reinforcement learning algorithms. It is unclear whether the new algorithm can still outperform the baselines upon increasing the complexity of the network architectures. In general, as the benchmark problem studied is simple, the claim that NEAT enables the new algorithm to solve challenging or complex problems is not strongly verified in the paper.
> >
> > Unfortunately, the subfield of open-ended learning is computationally expensive as it has to optimize for many environments. Similarly, EPOET tends to be a computationally expensive algorithm (120k CPU hours, as per our experiments) and the reason behind it is the fact that the algorithm keeps 20 active environment-agent pairs.
> >
> > As mentioned in the general comments, we have increased the number of runs to 4 seeds in our experimental results. This is in line with prior work who presented results based on 5 seeds. While ideally more runs would be better, the computational requirements of these open-ended methods make this infeasible.
> >
> > We mentioned in the paper that bigger networks may perform better than NEAT but then we can use algorithms that can scale to very deep neural networks, such as DeepNEAT [3].
> >
> > > The paper is well-written overall. However, several key concepts were used in the paper without thorough definition.
> >
> > Thank you very much. We have added more definitions in Appendix F.

---

> > > ### Author Response · Authors · 2022-11-18
> > > **Continuation of the comments**
> > >
> > > > The paper introduces a seemingly straightforward adoption of NEAT in an existing framework for open-ended learning. The novelty of the paper does not seem to be strong. The technical novelty and contribution of the newly developed transfer mechanisms also require more justifications.
> > >
> > > To clarify, we believe that the first novelty presented in the paper is the effect of having agents that can augment their topologies in an open-ended learning framework. As open-ended learning looks forward towards learning indefinitely, we provide insights on how augmentative topology agents provide stepping stones for completely open-ended algorithms. Here completely means to theoretically learn endlessly. We show this through the ANNECS score -- which is a measure of open-endedness, as provided by authors of EPOET [3]. Secondly, the novelty of the Species-Based Transfer mechanism is justified by providing the motivation that we have taken inspiration from the speciation property of the NEAT [4] algorithm to apply that for transfers of agents among environments. We have also empirically shown that the transfer mechanism is important for ATEP by showing that Random- and No Transfer-ATEP perform poorly.
> > >
> > > **References**
> > >
> > > [1] Brant, Jonathan C., and Kenneth O. Stanley. "Minimal criterion coevolution: a new approach to open-ended search." Proceedings of the Genetic and Evolutionary Computation Conference. 2017.
> > >
> > > [2] Stanley, Kenneth O. "Why open-endedness matters." Artificial life 25.3 (2019): 232-235.
> > >
> > > [3] Wang, Rui, et al. "Enhanced POET: Open-ended reinforcement learning through unbounded invention of learning challenges and their solutions." International Conference on Machine Learning. PMLR, 2020.
> > >
> > > [4] Stanley, Kenneth O., and Risto Miikkulainen. "Evolving neural networks through augmenting topologies." Evolutionary computation 10.2 (2002): 99-127.

---

> > > ### Comment · Reviewer_TYB9 · 2022-11-18
> > > **Thank you for your response**
> > >
> > > Thank the authors for responding to my concerns.
> > >
> > > I feel some responses are still not very convincing. For example, if open-ended learning is already computationally expensive, why should we further increase the computation complexity by adding extra functionality towards evolving networks with different architectures? The same concern also applies regarding the hyper-parameter settings. Meanwhile, agents with the capability of augmenting their topologies have been studied previously. I still don't quite understand why adopting such agents in the open-ended framework is challenging. The speciation idea explored in NEAT has also been utilized in NeuroEvoution and other evolutionary algorithms. I think the novelty of borrowing this idea for transfer learning needs stronger justifications.

---

> > > > ### Author Response · Authors · 2022-11-18
> > > > **Response for Clarification**
> > > >
> > > > We thank reviewer TYB9 again for the detailed reviews.
> > > >
> > > > > if open-ended learning is already computationally expensive, why should we further increase the computation complexity by adding extra functionality towards evolving networks with different architectures?
> > > >
> > > > While our method increases computational load, it also increases open-endedness of the algorithm which is measures by ANNECS score.
> > > >
> > > > While NEAT is computationally expensive, we can use other computationally inexpensive methods to increase the capacity and complexity of the network, such as Neurogenesis [1], as we suggested in conclusion and future works section 7. Thus the addition of extra functionality is due to the fact that currently EPOET exhibits performance plateaus which we don't want if we have to create an algorithm that can learn endlessly.
> > > >
> > > > > The same concern also applies regarding the hyper-parameter settings.
> > > >
> > > > We used a less deviated from standard NEAT hyper-parameters thus we performed very low hyper-parameter tuning.
> > > >
> > > > > Meanwhile, agents with the capability of augmenting their topologies have been studied previously. I still don't quite understand why adopting such agents in the open-ended framework is challenging.
> > > >
> > > > To clarify, our research is to show the community that augmentative topology agents are necessary for open-ended learning frameworks, and not adopting to these networks to open-ended learning framework. We showed that our goals were achieved by empirical results through improvement in ANNECS score and generalization abilities.
> > > >
> > > > > The speciation idea explored in NEAT has also been utilized in NeuroEvoution and other evolutionary algorithms. I think the novelty of borrowing this idea for transfer learning needs stronger justifications.
> > > >
> > > > Another clarification that we want to make is that as we refer to transfer mechanism in EPOET and ATEP, we do not mean the traditional transfer learning terminology used commonly in machine learning community. Here, transfer mechanism refers to transfer of agents between environments created by EPOET or ATEP in an attempt to solve the environment. In the Species-Based Transfer ATEP (SBT-ATEP) we use the speciation technique inspired from NEAT and look for best performing individuals (Neural networks) that are similar in network structure, within other Environment-Agent pairs. If we find similar ones we transfer the species of candidate best individual to the target best individuals environment and replace it with the candidate best individual's species. Similarity of networks can be calculated by equation 1 of Appendix D.  SBT-ATEP is further explained in the Section 5.1 of the paper.
> > > >
> > > > **References**
> > > >
> > > > [1] Timothy J Draelos, Nadine E Miner, Christopher C Lamb, Jonathan A Cox, Craig M Vineyard, Kristofor D Carlson, William M Severa, Conrad D James, and James B Aimone. Neurogenesis deep learning: Extending deep networks to accommodate new classes. In 2017 International Joint Conference on Neural Networks (IJCNN), pp. 526–533. IEEE, 2017.

---

### Official Review · Reviewer_JKxU · 2022-10-24

**Confidence:** 4
**Correctness:** 2
**Technical Novelty And Significance:** 3
**Empirical Novelty And Significance:** 3
**Recommendation:** 3

**Clarity, Quality, Novelty And Reproducibility:**

### Clarity
Several important details are unclear in the paper:
- The paper, while easy to follow at a high level, assumes much prior knowledge on some key concepts. The paper can benefit from providing intuitive and technical definitions of PATA-EC, ANNECS, and the novelty metric used by EPOET directly in the background section of the paper. These are core metrics that are never clearly defined.
- Similarly, a more technical definition of how FBT and SBT are performed would benefit clarity. A diagram illustrating examples of FBT and SBT would also be useful.
- A diagram of how NEAT can update a toy network would be useful to include in the background or method section.
- Equation 1 seems unnecessary for the main paper, and can instead be added to an appendix section detailing NEAT. This would free up room for including more details on the missing definitions described above. Further, the "excess and disjoint genes" are never defined, making this passage impossible to fully understand.
- Figure 1 should include references to the equation or section numbers where each component is defined, e.g. ANNECS.
- The key design choice for why FBT replaces the entire population upon transfer, rather than just some subset of individuals in the target environment population is unclear. SBT seems to benefit from this partial transfer, so it seems the FBT comparison is unfair in this regard.
- Figure 4 can benefit from more details in the caption about what the distributions on the top and right of the figure represent.
- The paper can benefit from stronger motivation for why ML research should care about open-endedness. The current discussion feels like a recitation of existing works without building a strong, compelling case for this line of work.
- I also suggest that authors restructure their paper to bring the contents of Appendix A into the main body of the paper, while improving the legibility and sizing of the plots in the paper.

### Quality
The paper feels quite rough in presentation, using even the wrong formatting for an ICLR paper, e.g. the horizontal margins are smaller than standard. The figures and captions are not polished and the overall writing can benefit from copy editing.

### Novelty
ATEP is a novel extension of POET and the main experimental results seem promising, despite the aforementioned issues in clarity of presentation and potential issues with reproducing the original EPOET results.

### Reproducibility
The paper provides enough details to reproduce the results in principle, but the release of their code (as promised in the paper) will greatly improve reproducibility.

**Strength And Weaknesses:**

### Strengths
- Open-endedness is an important, underexplored topic in ML research.
- The paper provides an interesting extension of EPOET that allows agent topologies to be co-evolved with the environment.
- The paper proposes two new transfer mechanisms for EPOET that are required for applying NEAT in updating the agent population, both of which serve as novel extensions to EPOET's standard transfer mechanism.

### Weaknesses
- It seems the comparison to EPOET is unfair because **NEAT requires ATEP to effectively maintain more agents than EPOET**. It is not clear if this paper controls for the larger population size of ATEP.
- The main experimental results (e.g. Figure 2, Figure 5) are only based on two training runs.
- The ANNECS values in Figure 2 for EPOET do not seem to match that from the original paper, where at 20k updates, EPOET reaches an ANNECS score of 100, thus matching the performance of SBT-ATEP in this work. This may be due to the authors reduction of the number of active environments from 40 to 20 with respect to the original study. However, the authors should run their implementation of EPOET on the original setting of 40 environments on 3 seeds to ensure their implementation is correct, or otherwise state that they reuse the official open source implementation of EPOET.
- The agent network size seems to play a significant role in improved ANNECS and robustness. Therefore, authors should also provide more detailed discussion on how the choice of 20x20 and 40x40 compares to the original EPOET implementation, which achieved much higher ANNECS scores in the same number of updates.
- The paper is unclear about several key details, described in the Clarity section.
- The paper claims the original POET work uses BipedalWalkerHardcore, but this is inaccurate. POET and EPOET used modifications of the BipedalWalker environment, not the specific BipedalWalkerHardcore environment, which refers to a specific setting of the BipedalWalker environment parameters.
- At the top of Page 6, the authors claim SBT-ATEP does not stagnate, but this does not follow from the results, because it is unknown if SBT-ATEP will stagnate beyond the number of training steps investigated.
- The paper does not provide the correct citation for the VAE. It should be Kingma, Diederik P., and Max Welling. "Auto-encoding variational bayes." 2013.
- The introduction mentions prior works are limited in open-endedness due to the finiteness of the environment. However, the authors do not state this shared weakness in their own experimental environment.
- The authors do not provide any measures of environment complexity over the course of training, e.g. as done in prior works. This would be useful for comparing the difference in kinds of environments produced by ATEP vs. EPOET.

**Summary Of The Paper:**

This paper provides an extension of the EPOET algorithm for co-evolving a population of RL (PPO) agents with a population of task configurations, such that the topology of agents can also be co-adapted throughout training. Specifically, this work introduces Augmentative Topology EPOET (ATEP), which uses NEAT instead of PPO to update the population of agents, and thus their topology in addition to network weights throughout training. As NEAT requires a population, ATEP effectively co-evolves populations of populations of agents, and thus requires new transfer mechanisms for transferring agents evolved to solve one environment to become the "elite" for another environment, as necessitated by EPOET. To this end, the paper proposes two such transfer mechanisms, Fitness-Based Transfer (FBT) and Species-Based Transfer (SBT). The main results show that ATEP methods result in more robust agent populations (i.e. capable of solving more task configurations) and leads to faster growth in the number of novel, meaningfully different tasks that are generated compared to baseline methods.

**Summary Of The Review:**

While I like the ideas in this paper, the overall evaluation is only over 2 training runs and the quality of the presentation and writing (including eliminating the existing incorrect claims previously detailed) can be much improved. Importantly, the authors do not provide reassurance that ATEP is not outperforming EPOET purely due to its use of a larger effective population size—since NEAT is a population-based approach and each agent in the population is now optimized according to a NEAT population. Further, I would like to see more evidence for the fidelity of their implementation of EPOET compared to the original, given the deviation in ANNECS score. For these reasons and those stated previously in my review, I cannot recommend this paper for acceptance in its current state. I am very open to accepting this paper if the authors can provide an improved version of this work that addresses the issues described.

---

> ### Author Response · Authors · 2022-11-18
> **Thank you for detailed reviews and addressing all comments**
>
> We would like to thank you reviewer JKxU for the detailed reviews. It will highly improve the paper and hopefully benefit the research community. Following are reviews with comments on how we addressed them. If the space limit for comments end, we will add extension in the comment.
>
> > This paper provides an extension of the EPOET algorithm for co-evolving a population of RL (PPO) agents with a population of task configurations, such that the topology of agents can also be co-adapted throughout training. Specifically, this work introduces Augmentative Topology EPOET (ATEP), which uses NEAT instead of PPO to update the population of agents, and thus their topology in addition to network weights throughout training. As NEAT requires a population, ATEP effectively co-evolves populations of populations of agents, and thus requires new transfer mechanisms for transferring agents evolved to solve one environment to become the "elite" for another environment, as necessitated by EPOET. To this end, the paper proposes two such transfer mechanisms, Fitness-Based Transfer (FBT) and Species-Based Transfer (SBT). The main results show that ATEP methods result in more robust agent populations (i.e. capable of solving more task configurations) and leads to faster growth in the number of novel, meaningfully different tasks that are generated compared to baseline methods.
>
> To clarify, EPOET uses Evolutionary Strategies (ES) rather than PPO for optimization although, in principle, it could use any RL algorithm.
>
> > It seems the comparison to EPOET is unfair because NEAT requires ATEP to effectively maintain more agents than EPOET. It is not clear if this paper controls for the larger population size of ATEP.
>
> We have now made it clear in the first paragraph of the Results and Discussion section (5) of the paper that we use the same number of max function evaluations ($10^{12}$) for each algorithm. We believe that having a larger population size is not unfair when an algorithm may only use the same amount of function evaluations. Here 1 function evaluation consists of 1 episode of the environment, containing a maximum of 2000 timesteps.
>
> > The main experimental results (e.g. Figure 2, Figure 5) are only based on two training runs.
>
> As mentioned in the general comment section, we have increased our experiments to 4 seeds. We would like to clarify that EPOET experimented on 5 seeds, presumably with significantly larger compute capacity. We experimented as much as we were able to and took our experiments closer to the number of their seeds.
>
> > The ANNECS values in Figure 2 for EPOET do not seem to match that from the original paper, where at 20k updates, EPOET reaches an ANNECS score of 100, thus matching the performance of SBT-ATEP in this work. This may be due to the authors reduction of the number of active environments from 40 to 20 with respect to the original study. However, the authors should run their implementation of EPOET on the original setting of 40 environments on 3 seeds to ensure their implementation is correct, or otherwise state that they reuse the official open source implementation of EPOET.
>
> The active environments were lessened from 40 to 20 to fit within our computational budget, as otherwise we would not have been able to perform experiments to the extent we did. This does affect the algorithm, in that the agents can use fewer iterations to optimize, as once we have exceeded the number of active environments, we archive the oldest environment-agent pair. This is consistent across all baselines and our algorithm. We have added more details in the paper to make the difference clear. We have used the original EPOET implementation for our baselines from https://github.com/uber-research/poet. We have added details regarding this comment in the Results and Discussion section (5), in the second paragraph.
>
> > The agent network size seems to play a significant role in improved ANNECS and robustness. Therefore, authors should also provide more detailed discussion on how the choice of 20x20 and 40x40 compares to the original EPOET implementation, which achieved much higher ANNECS scores in the same number of updates.
>
> EPOET40x40 has the original controller size (which we clarify in the paper). A higher ANNECS score in the original paper is because of having more active environments, thus more room for optimization per agent. We have updated the paper in the Experimental Section (4.3) by explaining why we chose EPOET20x20, and the reason is to have a smaller baseline to compare against.
>
> > The paper claims the original POET work uses BipedalWalkerHardcore, but this is inaccurate. POET and EPOET used modifications of the BipedalWalker environment, not the specific BipedalWalkerHardcore environment, which refers to a specific setting of the BipedalWalker environment parameters.
>
> Thank you for notifying this, we apologise for the mistake. We have updated it in the Enhanced POET section (3).

---

> > ### Author Response · Authors · 2022-11-18
> > **Continuation of the comments**
> >
> > > At the top of Page 6, the authors claim SBT-ATEP does not stagnate, but this does not follow from the results, because it is unknown if SBT-ATEP will stagnate beyond the number of training steps investigated.
> >
> > Thank you for notifying this. We have changed the statement to "We also find that SBT-ATEP has negligible performance plateaus during the run of our experiment in solving environments..".
> >
> > > The paper does not provide the correct citation for the VAE. It should be Kingma, Diederik P., and Max Welling. "Auto-encoding variational bayes." 2013.
> >
> > Thank you for the correction, we apologize for not having the right one. We have changed the citation to the one mentioned in the review.
> >
> > > The introduction mentions prior works are limited in open-endedness due to the finiteness of the environment. However, the authors do not state this shared weakness in their own experimental environment
> >
> > While we agree with the reviewer, this was not the focus of the work; thus it was only mentioned once in the conclusion and future works section.
> >
> > > The authors do not provide any measures of environment complexity over the course of training, e.g. as done in prior works. This would be useful for comparing the difference in kinds of environments produced by ATEP vs. EPOET
> >
> > While we agree with our reviewer that we have not provided the measure of environment complexity, we clarify that we use the same environment generation mechanism as EPOET, thus we know the complexity of the environment remains in the same statistical distribution.
> >
> > > The paper, while easy to follow at a high level, assumes much prior knowledge on some key concepts. The paper can benefit from providing intuitive and technical definitions of PATA-EC, ANNECS, and the novelty metric used by EPOET directly in the background section of the paper. These are core metrics that are never clearly defined.
> >
> > While we agree with the reviewer, we are limited with the space and would not be possible for us to add more details in the main body. We have added more details about PATA-EC and ANNECS score in Appendix F while the novelty metric is described in Section 5.1.
> >
> > > Similarly, a more technical definition of how FBT and SBT are performed would benefit clarity. A diagram illustrating examples of FBT and SBT would also be useful.
> >
> > Other than FBT- and SBT-ATEP explanations in the section 4.2, we added pseudocodes in the appendix as due to the page limitations we could not add further details.
> >
> > > A diagram of how NEAT can update a toy network would be useful to include in the background or method section.
> >
> > Thank you. We have added toy examples of mutation crossover operators in the Appendix E.1 and also some evolved networks from our experiments in Appendix E.2.
> >
> > > Equation 1 seems unnecessary for the main paper, and can instead be added to an appendix section detailing NEAT. This would free up room for including more details on the missing definitions described above. Further, the "excess and disjoint genes" are never defined, making this passage impossible to fully understand.
> >
> > We have added more details to excess and disjoint genes in the NEAT section 3 by adding "denoted excess or disjoint genes depending on whether it occurs within or outside the range of the other parent’s innovation numbers".
> >
> > > Figure 1 should include references to the equation or section numbers where each component is defined, e.g. ANNECS.
> >
> > Thank you for the advice. We have updated the caption with relevant references to the section where each component resides.
> >
> > > The key design choice for why FBT replaces the entire population upon transfer, rather than just some subset of individuals in the target environment population is unclear. SBT seems to benefit from this partial transfer, so it seems the FBT comparison is unfair in this regard.
> >
> > FBT is motivated by the exact same transfer mechanism presented by EPOET where it also transfers the whole population of ES individuals, making it a useful comparison. We have explained this more in the Section 5.1.
> >
> > > Figure 4 can benefit from more details in the caption about what the distributions on the top and right of the figure represent.
> >
> > We have added more description about the distribution in the caption of Figure 4.
> >
> > > The paper can benefit from stronger motivation for why ML research should care about open-endedness. The current discussion feels like a recitation of existing works without building a strong, compelling case for this line of work
> >
> > We have added more motivation in the introduction section by adding "Through the motivation by the fact that humans have always learnt and innovated in an open-ended manner, Open-ended learning research field emerged. For instance, humans did not invent microwaves to heat food, but to study radars. Vacuum tubes and electricity was invented for very different reason but we stumbled upon computers through them." [1]

---

> > > ### Author Response · Authors · 2022-11-18
> > > **Continuation of the comments**
> > >
> > > > I also suggest that authors restructure their paper to bring the contents of Appendix A into the main body of the paper, while improving the legibility and sizing of the plots in the paper.
> > >
> > > We respect the reviewer's comment but it is very difficult to adjust and bring more content to the main body as we are already at the edge of the limit.
> > >
> > > > The paper feels quite rough in presentation, using even the wrong formatting for an ICLR paper, e.g. the horizontal margins are smaller than standard. The figures and captions are not polished and the overall writing can benefit from copy editing.
> > >
> > > We apologize for the mistake, we have set the horizontal margins (an error that should not have occurred). We have polished the figures and captions while copy editing to the best we can.
> > >
> > > > ATEP is a novel extension of POET and the main experimental results seem promising, despite the aforementioned issues in clarity of presentation and potential issues with reproducing the original EPOET results.
> > >
> > > We appreciate the reviewer's detailed review. We hope that we give a impactful research to the community.
> > >
> > > > The paper provides enough details to reproduce the results in principle, but the release of their code (as promised in the paper) will greatly improve reproducibility.
> > >
> > > Thank you. For reviewers, the code is attached in the supplemental material and will be released as the paper gets published.
> > >
> > > **References**
> > >
> > > [1] Stanley, Kenneth O. "Why open-endedness matters." Artificial life 25.3 (2019): 232-235.

---

> ### Comment · Reviewer_JKxU · 2022-11-18
> **Thank you for the responses**
>
> I thank the authors for their responses to several of the concerns raised in my review. I appreciate the authors taking the time to update their manuscript with clarifications on many of these points.
>
> My main concern, and primary weakness stated in my review, still remains: It seems the improved performance of ATEP cannot be ruled out as being due to using a larger effective population of networks, due to NEAT. I do not think simply limiting the number of episodes for each method properly controls for this effect, since the benefits of a larger population may come from additional variability.
>
> One way to control for this would be to compare ATEP to a run of EPOET with an equivalently larger population size. Another, stronger baseline would be EPOET with the same effective population size as ATEP, but initialized to a random set of agent topologies that are *not* updated via NEAT over the course of training. These controls would provide stronger evidence for or against the benefits of NEAT as employed by ATEP.
>
> Given this remaining concern, I will keep my current rating for the paper.

---

> > ### Author Response · Authors · 2022-11-18
> > **Clarification on population size**
> >
> > We appreciate reviewers acknowledgment of our efforts.
> >
> > While we understand the reviewer's perspective, we would like to highlight Figure 2, which contains the ANNECS results. As is illustrated in the figure, SBT-ATEP  performed significantly better than both FBT-ATEP and EPOET40x40 (which have similar performance). This indicates that the population size is not the most important hyperparameter in EPOET's setting. We would also like to point out that EPOET40x40's performance is far superior to that of EPOET20x20, both of which had the same population size, which indicates that the size of the topology matters (which is not a surprising trend).
> >
> > In addition, if we consider Figure 4 and 5c, it is worth noting that SBT-ATEP almost reaches the same total number of nodes present in EPOET20x20, yet it performs significantly better, thus leading us to believe that the number of nodes, is not the sole contributing factor, but rather that the manner in which they are created (through NEAT in this case) is also a significant factor. Some complex topologies created by SBT-ATEP can be found in Appendix E of the revised paper.

---

> > > ### Comment · Reviewer_JKxU · 2022-11-19
> > > **Response to authors**
> > >
> > > I agree that the current results are promising, but would encourage the authors to run a well-controlled study with clear ablations to isolate which aspects of the method are contributing to the observed improvements over EPOET. It would also be useful to investigate why SBT leads to much smaller networks compared to FBT.
> > >
> > > After discussion with the authors, I plan to keep my current rating of the paper.

---

> > > > ### Author Response · Authors · 2022-11-23
> > > > **Thank you for the review**
> > > >
> > > > We thank our reviewer for the detailed review. As ATEP and EPOET are computationally expensive, a whole run with different population size would not be feasible for us but we will take this into account. We respect reviewers decision and hope that we can publish the research as it provides useful insights on why we should use augmentative topology agents in an open-ended learning framework.
> > > >
> > > > Thank you again!

---

### Official Review · Reviewer_TUdn · 2022-10-24

**Confidence:** 3
**Clarity, Quality, Novelty And Reproducibility:** 200000 CPU hours might be hard to rep…
**Correctness:** 2
**Technical Novelty And Significance:** 2
**Empirical Novelty And Significance:** 2
**Recommendation:** 3

**Strength And Weaknesses:**

1. the paper should be reorganized in the sake of readability.

For example "EPOET improves upon POET by adding in two algorithmic improvements" (1) ... (2).
What section does example (1) and (2)? If those the main contributions and improvements, one would expect to see them explicitly in the text.

2. a few relevant works are not mentioned. E.g., "Task-Agnostic Morphology Evolution" and similar.

3. I am not sure that a method can be called reproducible if it takes 200000 CPU hours. A simple examples clarifying the main points one by one will be helpful.

4. "Accumulated Number of Novel Environments Created and Solved (ANNECS), a metric for open-ended learning that, intuitively, describes the amount of interesting new...". What is the definition of "interesting"?
is Novel and interesting the same?



**Summary Of The Paper:**

this paper propose to simultaneously evolve the environments and the agents by increasing complexity of the later. the proposed method extends EPOET.

**Summary Of The Review:**

the lack of simple experiments demonstrating the key points and the lack of explicit definitions prevents me from recommending this paper for publication in ICLR.

---

> ### Author Response · Authors · 2022-11-18
> **Thank you for the detailed review and addressing all comments**
>
> We would like to thank you reviewer TUdn for the reviews. It will highly improve the paper and hopefully benefit the research community. Following are reviews with comments on how we addressed them.
>
> > the paper should be reorganized in the sake of readability. [Merging reviewers paragraph] For example "EPOET improves upon POET by adding in two algorithmic improvements" (1) ... (2). What section does example (1) and (2)? If those the main contributions and improvements, one would expect to see them explicitly in the text.
>
> We would like to clarify that because of the page limit constraints, we had to summarize the importance of these points in the main body. We have added further explanations in Appendix F.
>
> > a few relevant works are not mentioned. E.g., "Task-Agnostic Morphology Evolution" and similar.
>
> Thank you for the suggestion, we have added it in the related works section (2) and explained that the algorithm evolves morphologies without tasks, thus having the potential of learning open-endedly.
>
> > I am not sure that a method can be called reproducible if it takes 200000 CPU hours. A simple examples clarifying the main points one by one will be helpful.
>
> We agree with the reviewer that the method is computationally expensive. However, EPOET is also a computationally expensive (120000 CPU hours, per our experiments), yet reproducible algorithm. EPOET's results can be replicated, using their their own GitHub repository [https://github.com/uber-research/poet]. We would like to add that the Open-Ended learning domain is computationally expensive, similarly to many other domains in machine learning.
>
> > "Accumulated Number of Novel Environments Created and Solved (ANNECS), a metric for open-ended learning that, intuitively, describes the amount of interesting new...". What is the definition of "interesting"? is Novel and interesting the same?
>
> we have changed the wording accordingly and wrote "Accumulated Number of Novel Environments Created and Solved (ANNECS), a metric for open-ended learning that, intuitively, describes the amount of new content ..". To avoid confusion, we have removed reference the idea of "interesting" environments, and now solely refer to them as novel.
>
> > 200000 CPU hours might be hard to reproduce - simple set of experiment is required to understand the contributions.
> the main definitions/contributions are not formally provided (definition of interesting, etc)
>
> We agree with the reviewer that 200000 CPU hours are expensive to reproduce but this is a common case with the subfield as it tends to be computationally expensive. The main reason behind it is to optimize for many environments.
>
> We have omitted "interesting" for simplicity and we have edited the paper to use novelty only. We have added more definitions in Appendix E.

---

> > ### Comment · Reviewer_TUdn · 2022-11-18
> > **thanks for the clarifications**
> >
> > Dear Authors, how do you quantify novelty?

---

> > > ### Author Response · Authors · 2022-11-18
> > > **Thanks for the question**
> > >
> > > Dear reviewer, in particular case of EPOET and ATEP we use the rank-normalized PATA-EC score vector and apply euclidean distance on the vector. We have added more explanation of PATA-EC score in Appendix F and we explain the quantifying of novelty in Section 5.1.

---

### Official Review · Reviewer_SyVu · 2022-10-25

**Confidence:** 5
**Clarity, Quality, Novelty And Reproducibility:** 1. The paper is mostly readable excep…
**Correctness:** 2
**Technical Novelty And Significance:** 2
**Empirical Novelty And Significance:** 1
**Recommendation:** 3

**Strength And Weaknesses:**

1. The paper lacks sufficient insights/results/illustrations to understand how the agent topologies are changing, most of the methodology and evaluation is borrowed from EPOET whose focus was more on novel environment generation rather than agent topology, so the results and discussion seem inadequate. Further EPOET already introduced the idea of using NEAT for environment generation which the authors have adapted for topology evolution.

2. Statistical significance of the results is poor, only 2 seeds have been used for the comparisons. Performance of RL based agents is typically high variance, so it is not justified to compare the agent policies/performance with so less data.

3. Currently the work comes short on novelty and lacks reusable insights that can actually be informed for progressing open ended learning. Are there any interesting patterns in the way the topologies evolve? What insights can be drawn about choosing the hyperparameters for fitness and mutation for open ended learning?

4. To me, the empirical evaluation seems lacking as no adequate baseline for topology evolution is used.


**Summary Of The Paper:**

This paper proposes an ML pipeline to evolve agent topologies for open ended learning by combining previous works EPOET and NEAT. Experiments are done on the 2D bipedal environment.

**Summary Of The Review:**

Incremental idea with poor execution.

---

> ### Author Response · Authors · 2022-11-18
> **Thank you for the detailed review and addressing all comments.**
>
> We would like to thank you reviewer SyVu for the reviews. It will highly improve the paper and hopefully benefit the research community. Following are reviews with comments on how we addressed them. If the space reaches the limit we will add extension in the comment.
>
> > The paper lacks sufficient insights/results/illustrations to understand how the agent topologies are changing, most of the methodology and evaluation is borrowed from EPOET whose focus was more on novel environment generation rather than agent topology, so the results and discussion seem inadequate. Further EPOET already introduced the idea of using NEAT for environment generation which the authors have adapted for topology evolution.
>
> We have added an example of evolved topologies in the Appendix E for illustrations. We agree with the reviewer that EPOET's main focus is on novel environment generation but we would like to clarify that it is not solely the case. EPOET has to solve each environment that is created; thus the agents' performance becomes of utmost importance. We have seen in Figure 5 of the EPOET paper that there is a performance plateau for EPOET which is due to the agents' fixed topology by not having the capacity to solve complex environments [1]. This makes our contribution's main goal to prove that this performance plateau can be reduced by having agents that complexify over time.
>
> > Statistical significance of the results is poor, only 2 seeds have been used for the comparisons. Performance of RL based agents is typically high variance, so it is not justified to compare the agent policies/performance with so less data.
>
> As mentioned in the general comments, we have extended our experiments to 4 seeds (which is comparable to the 5 seeds used in the original EPOET paper). All results in the paper are updated accordingly. This limitation is due to the nature of the algorithm and competing methods in the subfield, which makes it computationally expensive.
>
> > Currently the work comes short on novelty and lacks reusable insights that can actually be informed for progressing open ended learning. Are there any interesting patterns in the way the topologies evolve? What insights can be drawn about choosing the hyperparameters for fitness and mutation for open ended learning?
>
>
> For more reusable insights, we have added examples of evolved topologies in Appendix E, alongside a discussion. Although we would like to emphasize that the main idea in our paper is not to discover insights into an agent's network structure. Rather, the main aim is that we should be able to relieve the human practitioner from the task of specifying a fixed neural network architecture by having an agent learn that for itself and grow in capacity as needed.
>
> > To me, the empirical evaluation seems lacking as no adequate baseline for topology evolution is used.
>
> We agree with our reviewer that there is no baseline for evolving topologies. The reason behind it is as our work's novelty is based on the evolution of topologies, and to the best of our knowledge, there is no work available that uses augmenting topologies in an open-ended framework. We do use two fixed-topology baselines derived from EPOET.
>
> > The paper is mostly readable except for the heavy use of abbreviations, the technical details about the existing methods used are not complete and can be improved by discussion in appendix for instance.
>
> We have added Appendix A for abbreviations to improve readability. For technical details we have all hyperparameter settings and pseudocodes available in the Appendix C and D, respectively. We have added more technical details about existing methods in Appendix F.
>
> > The paper lacks novel ideas and quality results as discussed above and would need significant improvements before being publishable.
>
> To summarize our main contributions: our novel idea is to explore the effects of having augmentative topology agents in an open-ended framework. We further introduced two novel transfer mechanisms. We empirically showed that these additions indeed improved the open-endedness (using the ANNECS score) and generalization capabilities (by experimenting on two generalization tests, results are illustrated in Figure 6) of EPOET.

---

> > ### Author Response · Authors · 2022-11-18
> > **Continuation of the comments**
> >
> > > Empirical evaluation is not adequate in terms of statistical significance, also on one domain is being used for comparison with the baseline.
> >
> > As mentioned earlier, we have increased our experiment to 4 seeds. The chosen domain is the canonical environment used to evaluate open-ended agents, as can be seen in the various related work [1,2,3]. This particular domain has numerous benefits, as discussed by EPOET's authors; specifically, the simple walking simulator, coupled with the large number of options for terrain makes it easy to observe and understand qualitatively different locomotion strategies simply by viewing them.
> >
> > Furthermore, the authors note that environments are easily modified, enabling numerous diverse terrain layouts to emerge to showcase the possibilities for adaptive specialization and generalization. Finally, it is relatively fast to simulate, allowing more iterations than a more complex environment would [1].
> >
> > **References**
> >
> > [1] Wang, Rui, et al. "Enhanced POET: Open-ended reinforcement learning through unbounded invention of learning challenges and their solutions." International Conference on Machine Learning. PMLR, 2020.
> >
> > [2] Wang, Rui, et al. "Paired open-ended trailblazer (poet): Endlessly generating increasingly complex and diverse learning environments and their solutions." arXiv preprint arXiv:1901.01753 (2019).
> >
> > [3] Parker-Holder, Jack, et al. "Evolving Curricula with Regret-Based Environment Design." arXiv preprint arXiv:2203.01302 (2022).

---

### Author Response · Authors · 2022-11-18
**Thank you all reviewers for comments and addressing a general comment.**

We thank all our reviewers for the detailed comments. We would like to address a general comment about the number of runs in the experiment. We agree that two runs is a low number. Unfortunately, the nature of methods in this subfield is such that they require large amounts of computation (for example, the work on EPOET [1] use only 5 seeds). Nonetheless, we have managed to double the number of runs reported in our experiments, bringing them in line with related work. The results are presented in the updated version of the paper, but do not change any of the analysis or takeaways.




[1] Wang, Rui, et al. "Enhanced POET: Open-ended reinforcement learning through unbounded invention of learning challenges and their solutions." International Conference on Machine Learning. PMLR, 2020.

---

### Comment · Area_Chair_ajqN · 2022-11-18
**Responses**

Dear Reviewers,

Do you have any comments/replies to author's responses - it would be great if you could respond to them. Have they changed your opinion on the paper?

Kind regards,
AC

---

### Decision · Program_Chairs · 2023-01-20

**Decision:**

Reject

**Justification For Why Not Higher Score:**

- There are problems with experimental evaluation, relating to number of runs, size of the agents, comparison given that we have a population size and reimplementation of the original method.
- The clarity and completeness of writing should be improved.
- The idea is too incremental without strong performance to justify it.

**Justification For Why Not Lower Score:**

N/A

**Metareview: Summary, Strengths And Weaknesses:**

The paper studies a co-evolution of agent topologies with environments. There are a number of issues raised by reviewers.
- There are problems with experimental evaluation, relating to number of runs, size of the agents, comparison given that we have a population size and reimplementation of the original method.
- The clarity and completeness of writing should be improved.
- The idea is too incremental without strong performance to justify it.